# Effects of Metabolizable Energy Intake and Body-Weight Restriction on Layer Pullets: 1-Growth, Uniformity, and Efficiency

Thiago L. Noetzold [1,*], Jo Ann Chew [1], Douglas R. Korver [1], René P. Kwakkel [1,2], Laura Star [3,4] and Martin J. Zuidhof [1]

1   Department of Agricultural, Food and Nutritional Science, University of Alberta, Edmonton, AB T6G 2P5, Canada; chew1@ualberta.ca (J.A.C.); dkorver@ualberta.ca (D.R.K.); rene.kwakkel@wur.nl (R.P.K.); mzuidhof@ualberta.ca (M.J.Z.)
2   Animal Nutrition Group, Department of Animal Sciences, Wageningen University, 6700 AH Wageningen, The Netherlands
3   Schothorst Feed Research, Meerkoetenweg 26, 8218 NA Lelystad, The Netherlands; lstar@schothorst.nl
4   Department of Applied Research, Aeres University of Applied Sciences, De Drieslag 4, 8251 JZ Dronten, The Netherlands
*   Correspondence: noetzold@ualberta.ca

**Abstract:** This study aimed to determine the effects of dietary energy and body-weight (BW) restriction on layer pullets' growth, uniformity, and feed efficiency. Two experiments were conducted using a precision feeding (PF) system (Experiment 1) and a conventional feeding (CON) system (Experiment 2). Experiment 1 consisted of a $2 \times 4$ factorial arrangement (eight treatments) with two feed allocation (FA) levels: meal every visit (MEV) or restricted to the lower boundary of Lohmann Brown-Lite pullets; and three dietary metabolizable energy (ME) levels: Low, Standard (Std), and High (2600, 2800, and 3000 kcal/kg, respectively); the fourth treatment enabled birds to choose from the three diets (Choice). Experiment 2 consisted of a $2 \times 3$ factorial arrangement (six treatments): two FA levels (*ad libitum* or restricted) and three dietary ME levels (Low, Std, and High). In each experiment, BW, coefficient of variation (CV), average daily feed intake (ADFI), average daily metabolizable energy intake (MEI), and feed conversion ratio (FCR) were recorded. Diet ADFI preferences and feeding motivation were determined only in the PF experiment. ANOVA was conducted on each experiment with the two main effects as fixed factors (FA and dietary ME), and age or period as the sources of variation. Differences were reported at $p \leq 0.05$. MEV (PF experiment) and *ad libitum*-fed (CON experiment) pullets had greater BW compared to restricted-fed pullets ($p < 0.05$). The lowest CV was observed in the restricted-fed pullets from the PF experiment ($p < 0.05$). ADFI was greater in pullets fed the Low ME diet in the PF experiment compared to all the other groups, and the lower the dietary ME, the greater the ADFI in the CON experiment ($p < 0.05$). Choice-feeding pullets preferred feed with greater ME content in the PF experiment ($p < 0.05$). The lower the dietary ME, the greater the FCR in the CON experiment ($p < 0.05$). Restricted-fed pullets had greater daily visits, and lower daily meals, meal size, and successful visits to the PF system ($p < 0.05$). In conclusion, the results of this trial indicated that lower dietary ME increased FCR and ADFI, whereas feed restriction decreased BW and increased feeding motivation. Future steps after this trial will include examining the effects of dietary energy and feed restriction on carcass composition and sexual maturation.

**Keywords:** dietary energy; feed restriction; laying hen; efficiency; smart agriculture; precision livestock feeding; caloric restriction

## 1. Introduction

Egg-type chickens are highly efficient in egg production due to advances in genetic selection. Several countries worldwide keep laying hens in production until they are up to 100 wks of age, with more than 500 eggs being produced per cycle [1,2]. Due to their enormous reproductive capacity, preparing pullets for an optimal laying cycle is critical [3].

Thus, the management and nutrition of modern laying-hens need to be constantly examined to meet this genetic potential and determine where improvements can be made in pullet nutrition and management.

Growth and development of a pullet plays an important role in the sexual maturation and reproductive performance of laying hens [4,5]. In the past, although there were differences among strains, pullets seemed to have a consistent minimal BW to commence sexual maturation within a given strain [6,7]. It is important to remember that birds can have similar BW but differ in their body composition, which can further increase or impair egg productivity in the flock. Many mechanisms regarding the ideal carcass composition and BW for optimal reproductive performance are still unclear [8].

As a means to change carcass composition and improve pullet growth and development, quantitative and qualitative feeding strategies can be implemented. In contrast with broiler breeders, feed restriction is not commonly implemented in commercial laying settings since these birds can autoregulate their feed intake, and in many instances insufficient feed intake can be a problem, such as in warm climate regions [9]. However, feed restriction programs might be helpful for optimal profit yield instead of optimal reproductive performance [10]. Restricted pullets have lower maintenance requirements, consuming less feed and being more efficient [11]. On the other hand, severe feed restriction during the rearing period can limit nutrient intake and decrease BW and egg production [12,13]. Additionally, restricted birds have increased lean mass deposition [14], which can further prevent birds from developing hemorrhagic liver syndrome [15], but also may decrease reproductive capacity.

Dietary energy has been extensively studied as a qualitative feed restriction strategy for optimal growth in broilers and egg production in hens [16–19]. Energy itself is not a nutrient but a property that nutrients can provide, mainly via carbohydrates, lipids, and proteins. Dietary energy dynamics in animals are complex depending on the dietary nutrients provided, and several reviews have shown their complexity in depth [20–22]. Ultimately, changes in dietary energy can enhance or reduce pullet growth rate and modify the growth of fat tissue, muscles and the relationship between fat and lean deposition [23,24].

In addition to the influence of BW and carcass composition on pullet performance, greater BW uniformity around the time of photostimulation is desirable. Flocks with lower uniformity will likely have more birds that respond differently to the same photostimulation and feed allowances [10], yielding poorer egg production over time [25]. The uniformity issue is more evident with broiler breeder hens that have a higher voracity and increased feeding competition. Nonetheless, factors such as nutrient availability, nutrient density, stocking density, feed delivery systems, and feeder space can affect the uniformity of laying pullets [5,26].

The objective of this study was to determine the effect of dietary metabolizable energy levels and restricted or non-restricted feeding trajectories (meal every visit in the PF experiment and *ad libitum* in the CON experiment) on BW, BW uniformity, ADFI, MEI, FCR, and feed motivation of laying pullets reared to 18 wks of age. It was hypothesized that pullets fed on every visit in the precision feeding experiment and *ad libitum* in the conventional experiment would have greater BW, ADFI, and MEI than restricted-fed pullets. Uniformity would decrease with restricted feeding in the CON experiment due to competition but increase with restricted feeding in the PF experiment due to an imposed upper BW limit. Additionally, FCR would decrease as birds consumed diets containing greater dietary ME; however, MEI would be similar across the different ME diets.

## 2. Materials and Methods

Two simultaneous experiments were conducted using a PF system (Experiment 1) and a conventional feeding system (Experiment 2). Due to the limited number of feeding stations, Experiment 2 was primarily used to complement Experiment 1 by providing additional birds for body composition and dissection evaluations. The PF system utilized in Experiment 1 was a second-generation precision feeding system developed at

the University of Alberta. The design and functionality are similar to the first generation of feeding stations [27]. The second-generation feeding stations were equipped with four independent feeders, enabling provision of treatment-specific diets to individual free-run birds. All procedures in these studies were approved by the University of Alberta Animal Care and Use Committee for Livestock and followed the Canadian Council on Animal Care guidelines [28].

### 2.1. Experimental Design and Diets

**Experiment 1.** The PF experiment consisted of 8 treatments in a 2 × 4 factorial arrangement with two feed allocation levels: meal every visit (MEV) or restricted; and four dietary apparent ME treatments (Low, Standard (Std), High, and Choice). The 3 levels of apparent dietary ME were 2600, 2800, and 3000 kcal/kg in the Low, Std, and High ME treatments, respectively. However, only one bird could eat at a time in the feeding stations, so access to feed was not *ad libitum* per se. In the Choice treatment, access to all diets was provided, and the birds chose which of the 3 feeds they wanted to consume. In the MEV feeding allocation group, birds were fed every time they went into the feeding stations. The MEV feeding group was intended to be analogous to the *ad libitum* group in the CON experiment. The experimental unit was the bird, with 23 birds per treatment combination, totaling one hundred and eighty-four Lohmann Brown-Lite one-day-old pullets at the time of placement.

**Experiment 2.** The conventional feeding experiment was conducted simultaneously with Experiment 1, primarily to evaluate the impact of dietary treatments on body composition. Experiment 2 consisted of 6 treatments in a 2 × 3 factorial arrangement: two levels of feed allocation (*ad libitum* or restricted) and three dietary apparent ME levels (Low, Std, and High). The experimental units were 2 replicate pens per treatment. There were 26 birds per replicate, totaling 312 Lohmann Brown-Lite one-day-old pullets in the CON experiment. There was no Choice feeding treatment in the CON experiment.

Feed was formulated based on the recommendations from the primary breeder [29] in a three-phase feeding program (starter, 1 to 48 d; developer, 49 to 111 d; pre-lay, 112 to 126 d) fed in mash form. All diets were formulated to have similar levels of crude protein, essential amino acids (AA), and crude fat (Table 1). Therefore, the apparent ME difference among the diets was modified by changing carbohydrate inclusion (starch vs. non-starch). The Std ME diet was made by blending the Low and High ME diets in a 1:1 ratio (Table 1).

**Table 1.** Dietary ingredients and calculated nutrient composition of experimental diets for Experiments 1 and 2 (%, as-fed basis).

| Ingredients | Starter (1 to 48 d) | | | Developer (49 to 111 d) | | | Pre-lay (112 to 126 d) | | |
|---|---|---|---|---|---|---|---|---|---|
| | Low ME | Std ME * | High ME | Low ME | Std ME * | High ME | Low ME | Std ME * | High ME |
| Wheat | 44.26 | | 60.51 | 49.04 | | 65.64 | 42.68 | | 59.37 |
| Soybean meal, 45% | 23.55 | | 19.21 | 13.80 | | 9.86 | 18.14 | | 14.33 |
| Oat hulls | 10.00 | | 6.60 | 10.00 | | 2.65 | 10.00 | | - |
| Cellulose | 8.58 | | - | 11.43 | | 6.25 | 6.17 | | - |
| Canola meal | 5.00 | | 5.00 | 7.25 | | 7.25 | 10.00 | | 10.00 |
| Canola oil | 4.59 | | 4.59 | 5.03 | | 5.03 | 5.49 | | 5.49 |
| Calcium carbonate | 1.45 | | 1.48 | 1.26 | | 1.29 | 5.27 | | 5.31 |
| Monocalcium Phosphate | 1.04 | | 0.99 | 0.55 | | 0.51 | 0.89 | | 0.85 |
| Vitamin and mineral premix [1] | 0.55 | | 0.55 | 0.55 | | 0.55 | 0.55 | | 0.55 |
| Choline Chloride [2] | 0.50 | | 0.50 | 0.50 | | 0.50 | 0.50 | | 0.50 |
| DL-Methionine, 99% | 0.14 | | 0.15 | 0.06 | | 0.06 | 0.04 | | 0.04 |
| Salt | 0.27 | | 0.25 | 0.22 | | 0.22 | 0.21 | | 0.21 |
| L-Lysine, 98.5% | 0.02 | | 0.10 | - | | - | - | | - |
| L-Threonine, 98.5% | - | | 0.02 | - | | - | - | | - |
| Phytase [3] | 0.01 | | 0.01 | 0.01 | | 0.01 | 0.01 | | 0.01 |
| Xylanase [4] | 0.05 | | 0.05 | 0.05 | | 0.05 | 0.05 | | 0.05 |
| Nutrient calculated (analyzed), % or as follows | | | | | | | | | |
| Metabolizable energy, kcal/kg | 2600 | 2800 | 3000 | 2600 | 2800 | 3000 | 2600 (2620) | 2800 (2879) | 3000 (3018) |

**Table 1.** *Cont.*

| Ingredients | Starter (1 to 48 d) | | | Developer (49 to 111 d) | | | Pre-lay (112 to 126 d) | | |
|---|---|---|---|---|---|---|---|---|---|
| | Low ME | Std ME * | High ME | Low ME | Std ME * | High ME | Low ME | Std ME * | High ME |
| Crude protein | 20.54 | 20.54 | 20.54 | 17.20 | 17.20 | 17.20 | 19.45 | 19.45 | 19.45 |
| Calcium | 1.05 | 1.05 | 1.05 | 0.90 | 0.90 | 0.90 | 2.50 | 2.50 | 2.50 |
| Available phosphorus | 0.45 | 0.45 | 0.45 | 0.37 | 0.37 | 0.37 | 0.45 | 0.45 | 0.45 |
| Lysine digestible | 0.98 | 0.98 | 0.98 | 0.75 | 0.72 | 0.70 | 0.89 | 0.87 | 0.84 |
| Methionine digestible | 0.40 | 0.40 | 0.40 | 0.28 | 0.28 | 0.28 | 0.29 | 0.29 | 0.29 |
| Methionine + Cystine digestible | 0.68 | 0.68 | 0.68 | 0.52 | 0.52 | 0.52 | 0.56 | 0.56 | 0.56 |
| Threonine digestible | 0.66 | 0.66 | 0.66 | 0.53 | 0.52 | 0.51 | 0.62 | 0.61 | 0.60 |

Analyzed values are presented between parentheses. * In all phases the Standard dietary ME was obtained by the mixture of Low and High ME diets in a 1:1 ratio. [1] Concentration per Kilogram of diet: 69 µg, 25-hydroxycholecalciferol (supplied at 0.05%, Rovimix HyD Premix, DSM Nutritional Products, Kaiseraugst, CH); vitamin A, 12.000 IU; vitamin E, 100 IU; vitamin C, 50 mg; vitamin $K_3$, 6 mg; vitamin B12, 35 ug; thiamine, 3 mg; riboflavin, 15 mg; vitamin B6, 6 mg; niacin, 40 mg; pantothenic acid, 25 mg; folic acid, 4 mg; biotin, 0.3 mg. Mn, 120 mg; Cu, 20 mg; Zn, 100 mg; Se, 0.30 mg; Fe, 80 mg; Iodine, 1.65 mg. [2] Provided 100 mg of choline per kg of diet. [3] Formulated to supply 500 FTU/kg *Escherichia coli*-derived phytase from *Trichoderma reesei* (Quantum Blue 5G, AB Vista Feed Ingredients, Marlborough, UK). [4] Avizyme 1302 Xylanase (5000 U/g) and protease (1600 U/g; Halchemix Canada Inc., Uxbridge, ON, Canada).

## 2.2. Birds and Management

Pullets in each of the PF and CON experiments were identified by a neck tag given at hatching, plus equipped with a radio frequency identification (RFID) wing tag at 14 d of age. Lighting and temperature management procedures during the rearing phase followed the breeder's recommendations [29]. Briefly, the photoschedule was 24L:0D for the first two days (25 lx), 16L:8D from days 3 to 5 (25 lx), and then maintained at 10L:14D (6 lx) until the end of this study portion (18 wks of age). Water was provided *ad libitum* in both experiments.

**Experiment 1.** Initially, pullets were placed in four 2.2 × 2.7 m pens containing one four-feeder feeding station each. Dietary ME treatments were initiated at placement where each station contained one of the four dietary energy treatments (Low, Std, High and Choice); pullets were placed in the pen that contained their corresponding dietary treatment group. The choice-feeding treatment pen had each diet randomly assigned to 3 feeders within each station, and the fourth feeder contained the Std ME diet. Pullets were trained to use the feeding stations during their first five weeks of life. Overall, the training period took 7 to 14 d [26], but due to the software changes required (this was the first experiment conducted with the four-feeder system), the training period was extended to 41 d. During the training period, birds in all treatments were allowed to consume feed at every visit. On day 42, individual precision feeding started, and birds were re-randomized among two large floor pens (4.5 × 5.4 m) with two four-feeder PF stations in each pen (approximately 45 birds per feeding station). The three dietary ME levels were randomly assigned to the four feeders of each feeding station. To ensure that all birds from any treatment could eat at any feeding station, at least one of each dietary ME option was available in each feeding station. Feeders were first randomly assigned to one of the three dietary ME levels (Low, Std, and High ME) and the fourth feeder had one of the three ME diets randomly assigned. The diet positions in the feeders were randomized in each feeding station. This dietary ME position randomization was conducted monthly to prevent bias due to feeder location preferences in the feeding station. Birds had access to only the feeds appropriate for their dietary treatment; the remaining feeders were closed. Birds on the Choice ME treatment had access to all feeders (all feeder doors opened). Feed allocation (MEV or restricted) started when the individual feeding started at 42 d of age.

Free-run birds were fed individually in the PF system. The system was composed of two stages, a pre-stage and a feeding stage. Once the bird entered the pre-stage, the system identified its RFID tag and, based on its treatment combination, the decision to feed or not was made. Birds from the MEV group were allowed to proceed to the feeding stage every time they entered the pre-stage, whereas birds from the restricted group were fed based on the lower limit of the Lohmann BW target recommended trajectory [29]. Thus, when birds

from the restricted BW trajectory treatment were above the target BW, they were gently ejected from the feeding station without access to a meal. The feeding stage was equipped with four feeders, and each feeder had a door, where the appropriate door opened for the dietary ME level of the given pullet that was allowed in the feeding stage. In the Choice feeding treatment, all four feeder doors were opened. All birds which qualified for a meal had access to the feed for 120 s.

**Experiment 2.** Pullets were placed in 12 small pens (1.75 × 1.8 m) with 26 birds per pen. Each pen had 1 pan feeder (33 cm diameter) providing 1.3 cm/bird of feeder space initially, which increased by 0.3 cm/bird every 4 wks after 10 wks of age due to birds being removed for dissections. *Ad libitum*-fed birds had free access to feed. Birds from the restricted group received feed daily, with allowances based on weekly pen average BW to achieve the lower boundary of the Lohmann BW target trajectory [29].

### 2.3. Data Collection

**Experiment 1.** Data were collected using the PF system. BW and feed intake (FI) were recorded at every feeding station visit. The FI for the training period (1 to 41 d) was calculated at the feeding station (pen) level and manual BW was recorded daily until 28 d. Median BW and total daily FI were calculated daily from the database recorded in the PF system. FCR calculations were conducted using the BW and FI records. MEI was calculated using the apparent metabolizable energy (AME; not corrected for nitrogen) results from Experiment 2. After the end of the training period, visit frequency, meal frequency, and meal size were calculated from the PF system database pooled as daily averages in four periods (6 to 8, 9 to 11, 12 to 14 and 15 to 18 wks of age). Successful visits to the feeding stations (effective meal percentage) were calculated as the daily meals divided by daily visits and the result multiplied by 100. BW uniformity was indirectly calculated using the coefficient of variation (CV), expressed as percentage of BW standard deviation divided by the average BW, and the result multiplied by 100.

**Experiment 2.** The BW of each bird was recorded manually on a weekly basis for CV and feed allocation calculations. Weekly FI was recorded at the pen level for FCR and MEI intake. MEI was calculated based on measured dietary AME in the pre-lay diet (not corrected for nitrogen retention). Two percent of acid insoluble ash marker (M; Celite 281, Lompoc, CA, USA) was mixed into the pre-lay diet and provided for 5 consecutive days. At 126 d, four birds per pen (eight per treatment) were randomly selected and euthanized for ileal content collection. Digesta samples were collected from Meckel's diverticulum to 3 cm before the ileal–cecal–colon junction of the intestinal tract, pooled within each pen and frozen at −20 °C. Samples were dried for 48 h at 60 °C and then ground for further analysis. Digestion of samples was performed with 4 N HCl and then the residues were ashed at 500 °C for 6 h [30]. Gross energy (GE) of feed and digesta samples was measured using bomb calorimetry (C200 Calorimeter, IKA Works Inc., Wilmington, NC, USA). The AME values were calculated as described by Kong and Adeola [31]: $AME = GE_{diet} - (GE_{digesta} \times (\frac{M_{diet}}{M_{digesta}}))$, where GE was the gross energy in the sample (kcal/kg), and M was the marker concentration in the sample. AME results are presented parenthetically in Table 1.

### 2.4. Statistical Analysis

All data were analyzed using the MIXED procedure in SAS (Version 9.4, SAS Institute Inc., Cary, NC, USA, 2012). Data from Experiment 1 were analyzed as a 2 × 4 factorial arrangement in a completely randomized design. Each bird was the replicate unit for treatment (23 replicates per treatment) for all parameters. Data from Experiment 2 were analyzed as a 2 × 3 factorial arrangement with a completely randomized design. Pen was the replicate unit for each treatment (2 pens per treatment combination) for ADFI, MEI, and FCR whereas for BW and CV, bird was the replicate unit, and the pen was included in the mixed model as a random source of variation. Time period (wks of age) was included in the model as a discrete source of the variation in each experiment for BW, CV, ADFI, and MEI.

Period was included as a source of variation for visit frequency, meal frequency, meal size and successful visits in the PF experiment. Individual birds were the subject to account for within-bird variation in the PF experiment. In the CON experiment, individual birds (BW and CV) or pens (ADFI, MEI, and FCR) were the subject to account for within-pen variation. Data from each experiment, linear and non-linear mixed models, were used to analyze differences among treatment means on all measured responses. Pairwise comparisons were used to estimate weekly significant differences for BW and CV in each experiment, by the PDIFF option of the LSMEANS statement and were reported as different when $p \leq 0.05$. A Tukey's range test was used to separate means, differences were reported when $p < 0.05$, and trends were reported where $0.05 \leq p < 0.10$.

## 3. Results and Discussion

### 3.1. Body Weight and Coefficient of Variation

Overall, BW was similar across the treatment combinations in the PF experiment from 0 to 11 wks of age (Table 2). In the PF experiment, pullets fed MEV had greater BW compared to restricted-fed pullets from 11 to 16 wks of age ($p < 0.05$). The interaction between the FA and Diet ME showed similar BW across the different dietary ME groups when pullets were restricted-fed, while 21 g lower BW average was observed in the High ME group compared to the Choice group when pullets were fed MEV ($p < 0.05$). The interaction between the FA and age showed increased BW in MEV-fed pullets as they aged compared to restricted-fed pullets ($p < 0.05$). No BW differences were observed in the CON experiment from 0 to 7 wks of age (Table 3). In the CON experiment, *ad libitum*-fed pullets had greater BW after 7 wks of age compared to restricted-fed pullets, which was maintained until the end of the rearing period ($p < 0.05$). The High ME group had a tendency of 15 g greater BW than the Low and Std ME ($p = 0.056$). The interaction between the FA and age showed increased BW in *ad libitum*-fed birds compared to restricted-fed pullets as they aged in the CON experiment ($p < 0.05$).

Egg-type pullets are known to have small BW difference when restricted-fed at 70% compared to *ad libitum* feeding by the time of photostimulation [32]. More recently, Bahry et al. [13] showed that Lohmann Brown-Lite pullets fed *ad libitum* had greater BW when compared to target-fed or those fed 20% below the target BW. In the above study, pullets were fed conventionally [13], similarly to the present study, where the CON experiment showed larger BW differences in the *ad libitum*- vs. restricted-fed birds compared to the MEV- vs. restricted-fed birds in the PF system. The smaller BW differences in the PF experiment compared to the CON experiment due to the FA factor is explained by the fact that MEV pullets still need to compete to access the feeding stations, while the corresponding feeding treatment birds in the CON experiment (*ad libitum*-fed pullets) had access to the feed at any time. On the other hand, when provided in an acceptable range (2600 to 3000 ME), dietary energy has been reported to have little influence on BW, since pullets tend to adjust their feed intake based on dietary ME concentration [33]. As shown in the current study, FA had greater influence over the BW response than the dietary ME factor as pullets aged.

**Table 2.** Effects of feed allocation and dietary metabolizable energy on BW during the rearing phase in Lohmann Brown-Lite pullets. Data from Experiment 1 (precision feeding).

| | Restricted | | | | | | | | Meal Every Visit | | | | | | | |
| --- | --- | --- | --- | --- | --- | --- | --- | --- | --- | --- | --- | --- | --- | --- | --- | --- |
| | Low ME | | Std ME | | High ME | | Choice | | Low ME | | Std ME | | High ME | | Choice | |
| Age (wks) | Mean | SEM | Mean | SEM | Mean | SEM | Mean | SEM | Mean | SEM | Mean | SEM | Mean | SEM | Mean | SEM |
| 0 (hatch) | 30.5 | 12.4 | 31.6 | 12.1 | 31.5 | 12.1 | 31.9 | 11.9 | 31.8 | 12.1 | 31.9 | 12.1 | 31.2 | 13.1 | 32.8 | 13.1 |
| 1 | 55.4 | 12.4 | 54.9 | 12.1 | 53.8 | 12.1 | 58.3 | 11.9 | 56.2 | 12.1 | 58.2 | 12.1 | 56.0 | 13.1 | 56.4 | 13.1 |
| 2 | 106 | 12.4 | 105 | 12.1 | 106 | 11.9 | 113 | 11.6 | 106 | 12.1 | 109 | 12.4 | 107 | 13.1 | 112 | 13.1 |
| 3 | 171 | 12.4 | 172 | 11.9 | 167 | 11.9 | 181 | 11.6 | 173 | 12.1 | 178 | 12.1 | 170 | 13.1 | 178 | 13.4 |
| 4 | 265 | 12.8 | 266 | 11.9 | 260 | 13.5 | 276 | 11.6 | 270 | 12.1 | 272 | 12.1 | 257 | 13.1 | 267 | 13.1 |
| 5 | 349 | 12.8 | 357 | 11.9 | 353 | 16.7 | 367 | 11.9 | 356 | 12.1 | 357 | 12.1 | 336 | 13.1 | 369 | 13.8 |
| 6 | 384 | 13.1 | 395 | 11.9 | 411 | 12.1 | 416 | 11.9 | 394 | 14.6 | 418 | 22.2 | 414 | 13.8 | 435 | 14.1 |
| 7 | 484 | 26.1 | 476 | 13.3 | 500 | 15.0 | 509 | 11.9 | 495 | 12.1 | 528 | 12.1 | 505 | 25.7 | 542 | 14.1 |
| 8 | 588 | 21.3 | 587 | 19.2 | 599 | 11.9 | 609 | 11.9 | 603 | 12.1 | 635 | 12.1 | 604 | 13.4 | 639 | 13.8 |
| 9 | 681 | 17.8 | 676 | 24.1 | 691 | 20.8 | 702 | 11.9 | 714 | 12.1 | 739 | 12.1 | 703 | 17.1 | 738 | 13.8 |
| 10 | 757 | 25.6 | 746 | 36.6 | 754 | 36.9 | 769 | 11.9 | 791 | 28.1 | 802 | 12.1 | 764 | 24.9 | 815 | 13.8 |
| 11 | 854 | 32.8 | 840 | 39.0 | 851 | 40.0 | 851 | 12.1 | 904 | 39.1 | 906 | 12.1 | 860 | 13.4 | 895 | 13.8 |
| 12 | 956 [ab] | 12.8 | 945 [b] | 14.2 | 953 [ab] | 13.2 | 947 [b] | 11.9 | 1024 [ab] | 19.9 | 1039 [a] | 14.8 | 1004 [ab] | 13.4 | 1025 [ab] | 14.1 |
| 13 | 1027 [ab] | 14.5 | 1014 [ab] | 26.3 | 1023 [ab] | 22.9 | 1022 [b] | 11.9 | 1099 [ab] | 41.3 | 1107 [a] | 12.1 | 1079 [ab] | 37.0 | 1108 [a] | 35.5 |
| 14 | 1089 [bc] | 12.8 | 1078 [c] | 14.1 | 1088 [c] | 11.9 | 1085 [c] | 11.9 | 1189 [ab] | 19.0 | 1185 [a] | 12.1 | 1155 [abc] | 23.0 | 1184 [a] | 15.4 |
| 15 | 1144 [b] | 12.8 | 1134 [bc] | 18.1 | 1150 [b] | 11.9 | 1132 [b] | 11.9 | 1251 [ab] | 26.3 | 1232 [ac] | 12.1 | 1218 [ab] | 30.3 | 1245 [a] | 30.5 |
| 16 | 1184 [bc] | 12.8 | 1177 [abc] | 20.6 | 1193 [abc] | 14.3 | 1180 [c] | 12.1 | 1272 [abc] | 33.0 | 1270 [a] | 12.4 | 1264 [abc] | 32.2 | 1267 [ab] | 34.5 |
| 17 | 1259 [bc] | 12.8 | 1244 [abc] | 19.1 | 1263 [bc] | 11.9 | 1254 [c] | 11.9 | 1356 [ab] | 19.5 | 1334 [ab] | 12.1 | 1333 [abc] | 34.8 | 1345 [a] | 33.8 |
| 18 | 1269 | 33.8 | 1268 | 20.6 | 1291 | 23.3 | 1285 | 11.9 | 1335 | 28.1 | 1319 | 19.2 | 1335 | 34.6 | 1342 | 39.4 |

| | *p*-value |
| --- | --- |
| FA [1] | <0.001 |
| Diet ME [2] | 0.56 |
| Age | <0.001 |
| FA × Diet ME | 0.011 |
| FA × Age | <0.001 |
| Diet ME × Age | 1.00 |
| FA × Diet ME × Age | 1.00 |

[a–c] Means within rows with no common subscript differ (*p* < 0.05). [1] Feed allocation (FA) at two levels: meal every visit and restricted to the lower boundary of the Lohmann Brown-Lite recommended target BW trajectory. [2] Dietary metabolizable energy treatments (Diet ME): Low (2600 kcal/kg), Standard (Std; 2800 kcal/kg), or High ME (3000 kcal/kg). Choice treatment enabled birds to choose from the three diets.

**Table 3.** Effects of feed allocation and dietary metabolizable energy on BW during the rearing phase in Lohmann Brown-Lite pullets. Data from Experiment 2 (conventional feeding).

| | Restricted | | | | | | Ad libitum | | | | | |
|---|---|---|---|---|---|---|---|---|---|---|---|---|
| | Low ME | | Std ME | | High ME | | Low ME | | Std ME | | High ME | |
| Age (wks) | Mean | SEM | Mean | SEM | Mean | SEM | Mean | SEM | Mean | SEM | Mean | SEM |
| | | | | | | g | | | | | | |
| 0 (hatch) | 31.4 | 10.0 | 31.2 | 9.9 | 30.9 | 9.8 | 30.7 | 10.0 | 31.6 | 9.9 | 32.6 | 9.7 |
| 1 | 55.5 | 10.0 | 58.6 | 9.9 | 60.2 | 9.8 | 58.1 | 10.0 | 58.0 | 9.9 | 60.4 | 9.7 |
| 2 | 101 | 9.9 | 109 | 9.9 | 107 | 9.8 | 107 | 9.9 | 105 | 9.9 | 108 | 9.7 |
| 3 | 171 | 9.9 | 175 | 9.9 | 173 | 9.9 | 174 | 9.9 | 172 | 10.0 | 174 | 9.7 |
| 4 | 255 | 9.9 | 260 | 9.9 | 255 | 9.8 | 258 | 9.9 | 254 | 10.0 | 258 | 9.8 |
| 5 | 361 | 9.9 | 370 | 9.9 | 357 | 9.8 | 364 | 9.9 | 360 | 10.0 | 364 | 9.8 |
| 6 | 444 | 9.9 | 445 | 9.9 | 461 | 9.8 | 492 | 9.8 | 487 | 10.0 | 494 | 9.7 |
| 7 | 519 [b] | 9.9 | 517 [b] | 9.9 | 544 [b] | 9.8 | 636 [a] | 17.2 | 629 [a] | 10.0 | 649 [a] | 9.7 |
| 8 | 653 [bc] | 13.5 | 641 [c] | 9.9 | 647 [bc] | 9.8 | 754 [ab] | 23.2 | 754 [a] | 10.0 | 790 [a] | 9.7 |
| 9 | 747 [b] | 9.9 | 741 [b] | 13.5 | 747 [b] | 9.8 | 883 [a] | 9.8 | 872 [a] | 11.5 | 894 [a] | 9.7 |
| 10 | 923 [ab] | 10.5 | 908 [b] | 10.0 | 915 [b] | 9.8 | 1018 [ab] | 29.1 | 1026 [a] | 21.2 | 1032 [ab] | 32.5 |
| 11 | 879 [b] | 10.8 | 891 [b] | 11.0 | 915 [b] | 10.7 | 1084 [a] | 21.6 | 1111 [a] | 11.0 | 1110 [a] | 23.1 |
| 12 | 992 [b] | 10.8 | 1002 [b] | 11.0 | 1025 [b] | 10.7 | 1187 [a] | 10.7 | 1216 [a] | 20.1 | 1210 [a] | 10.6 |
| 13 | 1071 [b] | 10.8 | 1093 [b] | 11.0 | 1100 [b] | 10.7 | 1268 [a] | 10.7 | 1278 [a] | 11.0 | 1278 [a] | 13.6 |
| 14 | 1228 [b] | 26.6 | 1239 [b] | 11.0 | 1258 [b] | 10.7 | 1363 [a] | 10.7 | 1373 [a] | 11.0 | 1379 [a] | 10.7 |
| 15 | 1208 [b] | 12.0 | 1204 [b] | 12.2 | 1226 [b] | 11.9 | 1372 [ab] | 36.8 | 1437 [a] | 18.6 | 1444 [a] | 11.7 |
| 16 | 1253 [b] | 12.0 | 1261 [b] | 12.2 | 1275 [b] | 11.9 | 1474 [a] | 13.2 | 1517 [a] | 12.2 | 1527 [a] | 11.7 |
| 17 | 1297 [b] | 13.8 | 1300 [b] | 12.2 | 1312 [b] | 11.9 | 1507 [a] | 11.9 | 1547 [a] | 12.2 | 1561 [a] | 13.4 |
| 18 | 1428 [b] | 30.4 | 1422 [b] | 17.4 | 1423 [b] | 21.0 | 1583 [a] | 11.9 | 1603 [a] | 16.3 | 1608 [a] | 11.7 |
| | | | | | | *p*-value | | | | | | |
| FA [1] | | | | | | <0.001 | | | | | | |
| Diet ME [2] | | | | | | 0.056 | | | | | | |
| Age | | | | | | <0.001 | | | | | | |
| FA × Diet ME | | | | | | 0.35 | | | | | | |
| FA × Age | | | | | | <0.001 | | | | | | |
| Diet ME × Age | | | | | | 0.68 | | | | | | |
| FA × Diet ME × Age | | | | | | 1.00 | | | | | | |

[a–c] Means within rows with no common subscript differ ($p < 0.05$). [1] Feed allocation (FA) at two levels: *ad libitum* and restricted to the lower boundary of the Lohmann Brown-Lite recommended target BW trajectory. [2] Dietary apparent metabolizable energy treatments (Diet ME): Low (2600 kcal/kg), Standard (Std; 2800 kcal/kg), or High ME (3000 kcal/kg).

As birds aged, the BW CV decreased in both trials ($p < 0.05$; Tables 4 and 5). In the PF experiment, the overall CV (Table 4) was lower in the restricted group over time compared to the MEV-fed pullets ($p < 0.05$). The average CV over the rearing period was lower in the High ME compared to the Std ME-fed pullets ($p < 0.05$; 7.3 vs. 8.5%). The interaction between the FA and age showed a more pronounced decrease in CV as birds aged for restricted-fed compared to the MEV-fed pullets ($p < 0.05$). In the CON experiment (Table 5), the interaction between the FA and dietary ME showed lower average CV in the rearing for restricted High ME-fed pullets compared to the Low and Std ME groups ($p < 0.05$; 8.7 vs. 10.7 and 10.3%, respectively), while that difference was not observed in the *ad libitum*-fed groups.

**Table 4.** Effect of feed allocation and dietary metabolizable energy on BW coefficient of variation (CV) during the rearing phase in Lohmann Brown-Lite pullets. Data from Experiment 1 (precision feeding).

| | Restricted | | | | | | | | Meal Every Visit | | | | | | | |
|---|---|---|---|---|---|---|---|---|---|---|---|---|---|---|---|---|
| | Low ME | | Std ME | | High ME | | Choice | | Low ME | | Std ME | | High ME | | Choice | |
| Age (wks) | Mean | SEM | Mean | SEM | Mean | SEM | Mean | SEM | Mean | SEM | Mean | SEM | Mean | SEM | Mean | SEM |
| | | | | | | | | % | | | | | | | | |
| 0 (hatch) | 8.6 ab | 3.8 | 6.3 b | 0.0 | 6.9 b | 0.1 | 9.3 ab | 2.9 | 7.8 ab | 0.7 | 8.0 a | 0.1 | 6.4 b | 0.0 | 6.9 ab | 3.9 |
| 1 | 12.0 ab | 2.8 | 15.5 ab | 3.3 | 16.2 a | 0.0 | 11.7 b | 0.6 | 12.9 ab | 4.2 | 11.5 ab | 3.8 | 12.6 ab | 2.9 | 16.2 a | 0.4 |
| 2 | 9.8 ab | 3.2 | 12.1 ab | 2.8 | 10.4 b | 0.1 | 10.0 ab | 1.3 | 14.6 ab | 3.3 | 9.2 ab | 3.8 | 11.7 a | 0.1 | 11.9 ab | 3.0 |
| 3 | 9.1 | 2.9 | 10.9 | 2.5 | 8.4 | 0.5 | 8.1 | 2.5 | 13.4 | 3.2 | 10.3 | 2.9 | 9.7 | 2.7 | 12.6 | 2.8 |
| 4 | 8.8 | 2.9 | 10.4 | 3.4 | 8.9 | 0.4 | 8.2 | 2.5 | 11.1 | 3.2 | 8.5 | 2.7 | 9.3 | 3.8 | 13.0 | 2.5 |
| 5 | 7.6 | 3.0 | 10.2 | 1.2 | 6.1 | 0.0 | 7.5 | 1.0 | 10.3 | 3.1 | 9.5 | 2.6 | 8.6 | 1.3 | 8.2 | 1.0 |
| 6 | 11.3 | 3.6 | 7.6 | 2.5 | 6.7 | 3.1 | 7.7 | 2.5 | 12.6 | 3.5 | 11.2 | 3.0 | 9.7 | 0.1 | 12.0 | 3.7 |
| 7 | 10.5 | 2.8 | 9.6 | 3.0 | 6.9 | 2.7 | 6.5 | 2.7 | 15.3 | 3.3 | 11.2 | 0.3 | 11.2 | 3.3 | 12.2 | 2.8 |
| 8 | 9.4 | 4.0 | 7.3 | 3.1 | 5.6 | 3.2 | 6.4 | 2.8 | 15.4 | 3.8 | 10.5 | 0.2 | 11.0 | 3.1 | 12.2 | 3.3 |
| 9 | 8.0 | 3.2 | 6.7 | 3.0 | 5.1 | 2.8 | 6.4 | 1.3 | 13.5 | 2.6 | 9.0 | 0.3 | 9.3 | 2.9 | 13.2 | 1.3 |
| 10 | 6.3 | 3.0 | 6.9 | 2.8 | 4.7 | 2.9 | 8.5 | 1.4 | 12.2 | 3.3 | 12.3 | 1.2 | 8.8 | 3.3 | 14.3 | 0.0 |
| 11 | 6.1 ab | 3.0 | 6.1 ab | 3.9 | 5.8 ab | 3.1 | 6.6 b | 0.3 | 11.2 ab | 2.7 | 10.0 a | 0.5 | 8.1 ab | 3.8 | 14.0 a | 1.0 |
| 12 | 3.5 abcd | 3.0 | 3.9 abcd | 3.8 | 2.3 d | 0.4 | 4.9 c | 0.1 | 10.0 abcd | 3.8 | 9.0 abc | 1.2 | 7.7 b | 0.2 | 11.1 a | 0.4 |
| 13 | 3.2 ab | 2.5 | 3.8 ab | 3.2 | 3.1 ab | 3.8 | 3.9 b | 0.0 | 9.5 ab | 2.6 | 9.9 a | 0.2 | 7.5 ab | 3.6 | 12.9 a | 1.3 |
| 14 | 2.3 ab | 3.6 | 3.0 ab | 3.7 | 2.4 b | 0.4 | 2.0 b | 0.5 | 8.1 ab | 3.8 | 9.2 a | 0.9 | 6.9 ab | 1.3 | 10.6 a | 0.6 |
| 15 | 1.7 c | 0.3 | 2.9 abc | 3.5 | 3.0 bc | 1.4 | 3.7 bc | 1.3 | 7.9 abc | 2.3 | 8.8 ab | 0.9 | 7.1 abc | 1.3 | 10.3 a | 0.6 |
| 16 | 1.8 | 1.7 | 4.1 | 3.6 | 2.6 | 1.2 | 4.6 | 1.3 | 8.0 | 3.5 | 9.3 | 1.2 | 6.5 | 0.7 | 9.0 | 1.0 |
| 17 | 1.7 c | 0.8 | 3.1 abc | 2.7 | 2.4 bc | 1.2 | 3.0 bc | 1.1 | 6.9 abc | 3.0 | 9.3 a | 0.5 | 6.5 abc | 0.8 | 7.9 ab | 0.3 |
| 18 | 2.9 ab | 1.8 | 5.1 ab | 3.8 | 3.3 b | 0.1 | 4.6 ab | 1.0 | 7.3 ab | 2.5 | 10.4 a | 5.8 | 7.5 a | 0.5 | 7.8 ab | 3.1 |

|  | *p*-value |
|---|---|
| FA [1] | <0.001 |
| Diet ME [2] | 0.004 |
| Age | <0.001 |
| FA × Diet ME | 0.12 |
| FA × Age | 0.034 |
| Diet ME × Age | 0.94 |
| FA × Diet ME × Age | 0.98 |

[a–d] Means within rows with no common subscript differ ($p < 0.05$). [1] Feed allocation (FA) at two levels: meal every visit and restricted to the lower boundary of the Lohmann Brown-Lite recommended target BW trajectory. [2] Dietary apparent metabolizable energy treatments (Diet ME): Low (2600 kcal/kg), Standard (Std; 2800 kcal/kg), or High ME (3000 kcal/kg). Choice treatment enabled birds to choose from the three diets.

**Table 5.** Effect of feed allocation and dietary metabolizable energy on BW coefficient of variation (CV) during the rearing phase in Lohmann Brown-Lite pullets. Data from Experiment 2 (conventional feeding).

| | Restricted | | | | | | Ad libitum | | | | | |
|---|---|---|---|---|---|---|---|---|---|---|---|---|
| | Low ME | | Std ME | | High ME | | Low ME | | Std ME | | High ME | |
| Age (wks) | Mean | SEM | Mean | SEM | Mean | SEM | Mean | SEM | Mean | SEM | Mean | SEM |
| | | | | | | | % | | | | | |
| 0 (hatch) | 8.4 | 2.0 | 7.5 | 0.0 | 8.2 | 0.9 | 8.7 | 0.5 | 7.6 | 1.3 | 8.4 | 1.3 |
| 1 | 15.2 | 1.8 | 12.3 | 1.1 | 10.6 | 1.0 | 13.6 | 1.4 | 11.8 | 0.3 | 10.8 | 1.2 |
| 2 | 13.8 | 1.6 | 10.5 | 1.6 | 10.6 | 0.4 | 11.3 | 1.4 | 10.3 | 0.1 | 9.7 | 1.1 |
| 3 | 12.9 | 1.3 | 10.7 | 0.8 | 10.7 | 1.0 | 10.2 | 0.9 | 9.2 | 1.1 | 8.9 | 0.1 |
| 4 | 12.2 | 1.2 | 10.0 | 1.2 | 11.0 | 0.5 | 9.6 | 0.4 | 8.6 | 1.3 | 9.1 | 0.9 |
| 5 | 11.1 | 0.6 | 9.8 | 0.5 | 11.0 | 0.8 | 9.6 | 0.7 | 8.5 | 1.1 | 8.8 | 1.1 |
| 6 | 11.1 [ab] | 0.9 | 10.6 [ab] | 1.3 | 10.4 [a] | 0.2 | 9.2 [ab] | 0.2 | 8.9 [b] | 0.0 | 9.0 [ab] | 1.6 |
| 7 | 12.1 [ab] | 0.7 | 11.7 [a] | 0.0 | 9.9 [bc] | 0.3 | 8.0 [c] | 0.6 | 8.9 [bc] | 0.2 | 8.8 [abc] | 1.1 |
| 8 | 12.0 [ab] | 1.3 | 11.6 [ab] | 0.8 | 9.8 [a] | 0.1 | 8.1 [b] | 0.1 | 8.8 [b] | 0.0 | 8.3 [ab] | 0.8 |
| 9 | 11.1 [abc] | 0.9 | 10.2 [abc] | 2.2 | 10.1 [a] | 0.1 | 7.1 [c] | 0.3 | 9.2 [b] | 0.1 | 8.4 [c] | 0.2 |
| 10 | 10.5 | 0.9 | 10.5 | 1.7 | 9.1 | 0.0 | 6.4 | 0.7 | 9.1 | 0.1 | 8.2 | 0.3 |
| 11 | 10.7 [ab] | 1.7 | 12.9 [a] | 0.9 | 8.2 [ab] | 0.8 | 6.9 [ab] | 1.3 | 8.7 [ab] | 0.8 | 8.3 [b] | 0.2 |
| 12 | 10.8 [abc] | 1.3 | 12.1 [a] | 0.1 | 7.7 [bc] | 0.9 | 5.7 [c] | 0.1 | 8.7 [b] | 0.5 | 9.1 [b] | 0.2 |
| 13 | 9.5 [a] | 0.0 | 10.9 [a] | 1.0 | 7.2 [ab] | 1.1 | 5.9 [b] | 0.4 | 7.2 [ab] | 0.9 | 8.1 [ab] | 0.7 |
| 14 | 9.0 [a] | 0.1 | 9.6 [abc] | 1.2 | 6.0 [c] | 0.0 | 5.9 [bc] | 0.2 | 6.7 [bc] | 0.4 | 7.4 [b] | 0.3 |
| 15 | 8.6 | 1.6 | 9.9 | 1.4 | 6.3 | 1.5 | 5.8 | 0.0 | 6.8 | 1.2 | 7.0 | 0.4 |
| 16 | 7.9 | 0.3 | 9.4 | 0.7 | 6.5 | 1.4 | 6.1 | 0.5 | 6.7 | 1.7 | 6.6 | 0.8 |
| 17 | 8.2 | 1.1 | 8.5 | 2.0 | 5.9 | 1.1 | 5.2 | 0.3 | 7.8 | 1.4 | 6.4 | 1.4 |
| 18 | 8.1 [a] | 0.2 | 7.8 [ab] | 1.4 | 5.8 [b] | 0.2 | 5.9 [ab] | 1.1 | 7.8 [ab] | 1.7 | 7.1 [ab] | 0.9 |
| | | | | | | | p-value | | | | | |
| FA [1] | | | | | | | <0.001 | | | | | |
| Diet ME [2] | | | | | | | <0.001 | | | | | |
| Age | | | | | | | <0.001 | | | | | |
| FA × Diet ME | | | | | | | <0.001 | | | | | |
| FA × Age | | | | | | | 0.47 | | | | | |
| Diet ME × Age | | | | | | | 0.091 | | | | | |
| FA × Diet ME × Age | | | | | | | 0.13 | | | | | |

[a–c] Means within rows with no common subscript differ ($p < 0.05$). [1] Feed allocation (FA) at two levels: *ad libitum* and restricted to the lower boundary of the Lohmann Brown-Lite recommended target BW trajectory. [2] Dietary metabolizable energy treatments (Diet ME): Low (2600 kcal/kg), Standard (Std; 2800 kcal/kg), or High ME (3000 kcal/kg).

Body weight coefficient of variation is a measure of the BW variability, where the lower the CV, the higher the BW uniformity. Substantial increases in uniformity have been reported in broiler breeder hens fed in the PF system when compared to conventionally fed birds, with CV of less than 2% around the time of photostimulation [26,27]. In the present study, the smallest CV at 18 wks of age was observed in the restricted-fed pullets from the PF experiment due to the fact that birds were restricted to an upper BW target.

*3.2. Feed and Metabolizable Energy Intake*

In each of the PF and CON experiments, MEV- and *ad libitum*-fed pullets had greater FI than restricted-fed pullets, respectively ($p < 0.05$; Table 6). In the PF experiment, ADFI was greater in pullets fed a Low ME diet compared to Std, High and Choice treatments, whereas in the CON experiment the lower the dietary ME, the greater the ADFI (Low ME > Std ME > High ME; $p < 0.05$; Table 6). In the CON experiment, FI in the FA treatment depended on dietary ME treatment. Pullets fed *ad libitum* had similar ADFI in the Low and Std ME levels, but a wider ADFI difference in all three dietary ME groups was observed when feed restricted ($p < 0.05$). As expected, FI increased as age increased in the PF and CON experiments ($p < 0.05$). Interactions between the FA and age, and dietary ME and age were present in both experiments ($p < 0.05$; Table 6). In the PF and CON experiments, as pullets aged, ADFI increased in MEV- and *ad libitum*-fed pullets compared to the restricted birds, as well as in birds fed Low ME energy levels compared to the other energy levels.

**Table 6.** Average daily feed intake (ADFI) and average daily metabolizable energy intake (MEI) of Lohmann Brown-Lite pullets using different feed allocation (FA) and dietary metabolizable energy levels (Diet ME) in Experiment 1 (precision feeding; PF) and Experiment 2 (conventional; CON), 0 to 18 wks of age.

| Effect | | PF Experiment [1] | | | | CON Experiment [2] | | | |
|---|---|---|---|---|---|---|---|---|---|
| | | ADFI | | MEI | | ADFI | | MEI | |
| | | Mean | SEM | Mean | SEM | Mean | SEM | Mean | SEM |
| | | —— g —— | | —- kcal/d —- | | —— g —— | | —- kcal/d —- | |
| FA | Restricted | 41.1 [b] | 0.2 | 116.5 [b] | 0.6 | 48.8 [b] | 0.2 | 138.2 [b] | 0.5 |
| | MEV or *ad libitum* | 43.0 [a] | 0.3 | 121.8 [a] | 1.0 | 59.6 [a] | 0.3 | 168.8 [a] | 0.9 |
| Diet ME | Low ME | 45.0 [a] | 0.4 | 117.8 [b] | 1.1 | 57.2 [a] | 0.4 | 149.9 [b] | 1.0 |
| | Std ME | 41.5 [b] | 0.5 | 119.4 [ab] | 1.5 | 54.7 [b] | 0.3 | 157.6 [a] | 0.8 |
| | High ME | 40.4 [b] | 0.3 | 122.0 [a] | 1.0 | 50.7 [c] | 0.4 | 152.9 [b] | 0.8 |
| | Choice | 41.4 [b] | 0.4 | 117.3 [b] | 1.0 | - | - | - | - |
| FA × Diet ME | Low ME × Restricted | 44.2 | 0.5 | 115.9 | 1.3 | 51.7 [c] | 0.4 | 135.5 [d] | 0.9 |
| | Std ME × Restricted | 40.4 | 0.5 | 116.3 | 1.4 | 48.7 [d] | 0.2 | 140.3 [c] | 0.9 |
| | High ME × Restricted | 39.6 | 0.4 | 119.4 | 1.2 | 46.0 [e] | 0.4 | 138.8 [cd] | 0.7 |
| | Choice × Restricted | 40.3 | 0.5 | 114.5 | 1.2 | - | - | - | - |
| | Low ME × MEV or *ad libitum* | 45.7 | 0.7 | 119.7 | 1.8 | 62.7 [a] | 0.7 | 164.4 [b] | 1.8 |
| | Std ME × MEV or *ad libitum* | 42.6 | 0.9 | 122.5 | 2.6 | 60.8 [a] | 0.4 | 175.0 [a] | 1.4 |
| | High ME × MEV or *ad libitum* | 41.3 | 0.5 | 124.6 | 1.6 | 55.3 [b] | 0.6 | 167.0 [b] | 1.4 |
| | Choice × MEV | 42.3 | 0.5 | 120.2 | 1.6 | - | - | - | - |
| | | *p*-value | | | | | | | |
| FA | | <0.001 | | <0.001 | | <0.001 | | <0.001 | |
| Diet ME | | <0.001 | | 0.004 | | <0.001 | | <0.001 | |
| Age | | <0.001 | | <0.001 | | <0.001 | | <0.001 | |
| FA × Diet ME | | 0.94 | | 0.90 | | 0.010 | | 0.013 | |
| FA × Age | | <0.001 | | <0.001 | | <0.001 | | <0.001 | |
| Diet ME × Age | | <0.001 | | <0.001 | | <0.001 | | <0.001 | |
| FA × Diet ME × Age | | 0.99 | | 0.99 | | 0.34 | | 0.24 | |

[a–e] Means within columns with no common subscript differ ($p < 0.05$). [1] Feed allocation (FA) at two levels: meal every visit (MEV) and restricted to the lower boundary of the Lohmann Brown-Lite recommended target BW trajectory. Dietary metabolizable energy treatments (Diet ME): Low (2600 kcal/kg), Standard (Std; 2800 kcal/kg), or High ME (3000 kcal/kg). Choice treatment enabled birds to choose from the three diets. [2] Feed allocation (FA) at two levels: *ad libitum* and restricted to the lower boundary of the Lohmann Brown-Lite recommended target BW trajectory. Dietary metabolizable energy treatments (Diet ME): Low (2600 kcal/kg), Standard (Std; 2800 kcal/kg), or High ME (3000 kcal/kg).

In the PF and CON experiments, restricted-fed birds had lower MEI compared to the MEV and *ad libitum*-fed birds ($p < 0.05$; Table 6). Average daily MEI increased in pullets fed the High ME diets compared to the Low ME and Choice feeding groups in the PF experiment, while in the CON experiment, the greater MEI was observed in birds fed the Std ME compared to the Low and High ME levels ($p < 0.05$). Additionally, the interaction between feed allocation and dietary ME in the CON experiment showed that restricted-fed pullets had similar MEI between Std and High ME diets, but this difference was not observed when birds were fed *ad libitum* ($p < 0.05$). MEI intake increased as age increased in each experiment as expected ($p < 0.05$). In the PF experiment, the interactions between the FA and age showed that as pullets aged, MEI increased in the MEV group from 12 to 15 wks of age when compared to the restricted-fed birds ($p < 0.05$; Figure 1A). Similarly, the interaction between the FA and age in the CON experiment showed increased MEI from 6 to 18 wks of age in the *ad libitum*-fed pullets compared to restricted pullets ($p < 0.05$). Dietary ME and age interaction demonstrated that pullets fed High ME levels had a greater MEI intake at 15 wks of age compared to the Choice group, and greater MEI from 17 to 18 wks of age compared to the Low ME in the PF experiment ($p < 0.05$; Figure 1B). In the CON experiment, the interaction between dietary ME and age showed that pullets fed High ME levels had a greater MEI intake from 6 to 7 wks of age compared to the Low ME group, while at 17 wks of age, the greater MEI was in the Std ME group compared to the High ME diet group ($p < 0.05$).

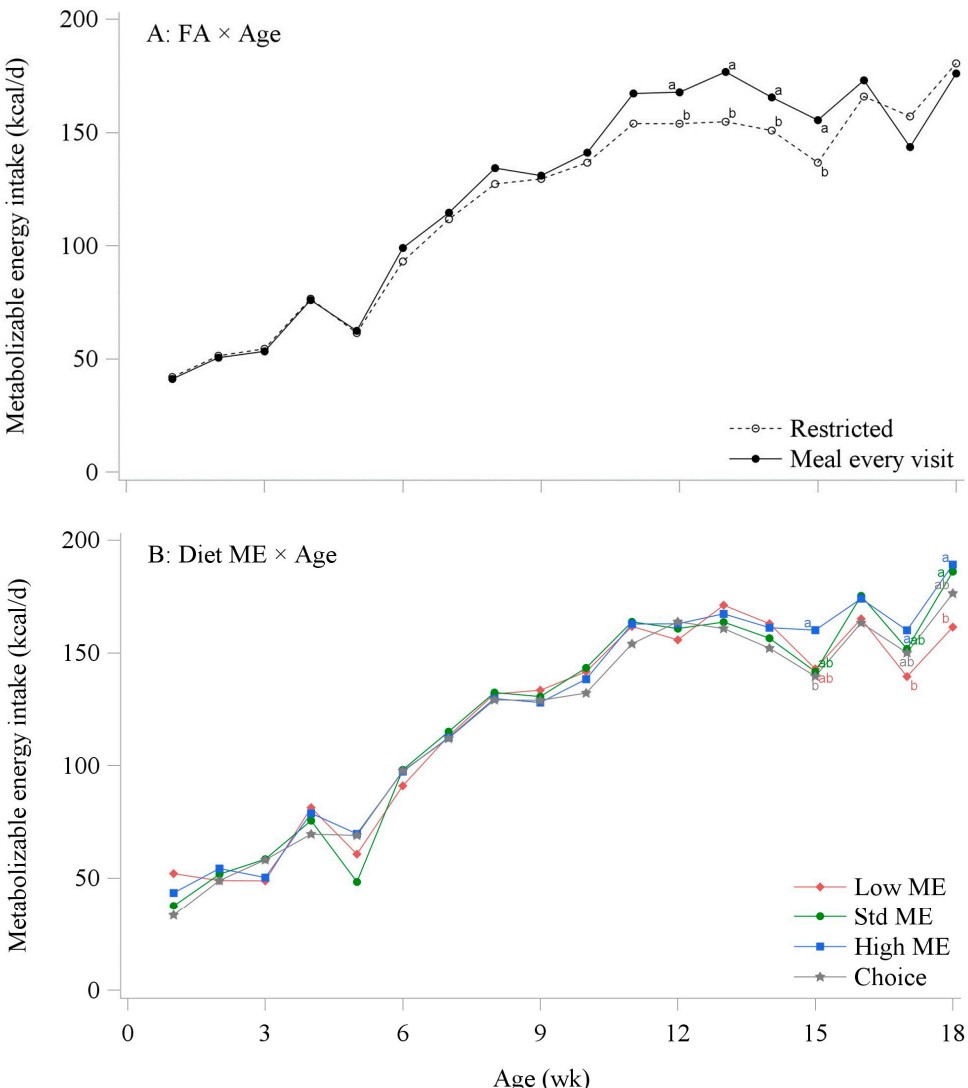

**Figure 1.** Average metabolizable energy intake (kcal/d) of Lohmann Brown-Lite pullets. Interaction effects between feed allocation (FA) and age (**A**), and dietary metabolizable energy (Diet ME) and age (**B**). Data from Experiment 1 (precision feeding). Different letters between treatments indicate significant differences at *p* < 0.05.

Although birds fed low dietary energy levels tended to increase feed intake to account for their energy requirements [34], in the present study the increased ADFI from pullets in the Low ME group was insufficient to maintain the same MEI as the other dietary ME treatments, potentially because feed intake capacity might have reached a limit and prevented birds from higher feed consumption. Similar responses have been observed before when feeding diets containing 2600 vs. 3100 kcal/kg [35]. However, smaller ranges in dietary ME levels affected ADFI but not overall MEI because of the ability of birds to adjust their FI [36,37]. Additionally, when unbalanced diets were provided (e.g., different energy level or protein ratio), there was some evidence that birds adjusted their feed intake in an attempt to compensate for dietary factors other than energy [38], which might explain the greater ME intake from Std ME birds in the CON experiment. The AA levels as a ratio to dietary energy were high and low in the Low and High ME diets, respectively. Consequently, the AA intake increased based on FI and was contrary to the dietary energy level. This might suggest that when using the ideal protein concept, the dietary energy intake is prioritized over AA intake despite the different energy to protein ratio in the diets.

Feed intake control in chickens is complex, and several factors can play a role in its regulation, such as dietary nutrient composition [39]. While results from some research

support the ability of birds to regulate their feed consumption to meet energy requirements in broilers, laying hens and broiler breeders [17,40,41], other studies have shown no effect on feed intake due to dietary energy in chickens [24,36,42]. Several effects might not be accounted for in the above studies, such as age, breed, environment, dietary protein level, and dietary fat, which are the main confounding factors related to the energy requirements [39]. In the present study, dietary energy was manipulated only by altering carbohydrate inclusion levels, maintaining similar levels of crude fat, crude protein, and essential AA.

In the PF experiment, the Choice feeding treatment diet preferences were explored. Pullets preferred feed with greater ME content, where the High ME diet had the greatest intake preference and the Low ME diet had the lowest ($p < 0.05$; Table 7). This feed consumption reflected a similar ME intake pattern to the different ME diets in the Choice group ($p < 0.05$; Table 7). It has been reported that when given a choice, birds can select one diet over another with different dietary protein and ME levels [43,44]. Nutrient intake self-regulation in animals is usually associated with physiological requirements and feed nutrient availability [45]. Furthermore, feed intake preferences could be associated with feed palatability [46]. Crude fat is reported to increase feed intake due to palatability even when the dietary energy is kept constant [47]. Interestingly, dietary cellulose is reported to decrease the initial feed intake preference (24 h) due to diet palatability in White Leghorn pullets [48]. Because in the current experiment crude fat was maintained similar across diets, and cellulose level was the main shift among the dietary ME groups, the perception that cellulose has a palatability effect could be hypothesized. However, when birds were not given a choice, their FI increased in the long-term rearing period regardless of the cellulose level. This might suggest that the intake was driven by energy requirements rather than palatability. Additionally, in performance studies, it is difficult to separate the palatability effect from nutrient driven effects [49]. Although we cannot completely attribute the greater dietary ME intake preference to one of the factors, in the current trial, the preference for consuming higher energy diets might have been driven by energy requirements, where energy requirements of the birds would be met with less feed intake when consuming diets with greater ME.

**Table 7.** Daily feed intake preference (ADFI) and metabolizable energy intake (MEI) by dietary energy of Lohmann Brown-Lite pullets in the Choice feeding treatment. Data from Experiment 1 (precision feeding), 0 to 18 wks of age.

| FA [1] | Diet Option [2] | ADFI | | MEI | |
|---|---|---|---|---|---|
| | | Mean | SEM | Mean | SEM |
| | | —— g/diet/d —— | | —— kcal/diet/d —— | |
| | Low ME | 11.5 [c] | 0.7 | 30.6 [c] | 2.0 |
| | Std ME | 15.1 [b] | 0.9 | 44.8 [b] | 2.7 |
| | High ME | 18.1 [a] | 0.8 | 56.6 [a] | 2.3 |
| | *Total intake* | *44.7* | | *132* | |
| Restricted | Low ME | 11.4 [c] | 1.1 | 30.8 [cd] | 2.9 |
| | Std ME | 14.7 [bc] | 1.0 | 42.4 [bc] | 2.9 |
| | High ME | 18.7 [a] | 1.0 | 57.2 [a] | 3.0 |
| Meal every visit | Low ME | 11.5 [c] | 0.9 | 30.3 [d] | 2.8 |
| | Std ME | 16.1 [abc] | 1.5 | 47.2 [ab] | 4.5 |
| | High ME | 18.0 [ab] | 1.1 | 56.1 [a] | 3.4 |
| | | *p*-value | | | |
| Diet option | | <0.001 | | <0.001 | |
| FA | | 0.82 | | 0.73 | |
| FA × Diet option | | 0.45 | | 0.66 | |

[a–d] Means within columns with no common subscript differ ($p < 0.05$). [1] Feed allocation at two levels: meal every visit and restricted to the lower boundary of the Lohmann Brown-Lite recommended target BW trajectory. [2] Choice treatment enabled birds to choose from the three dietary apparent metabolizable energy levels.

### 3.3. Feed Conversion Ratio

In the PF experiment, FA levels did not affect cumulative FCR ($p > 0.05$; Table 8). However, birds fed the Low ME diet had greater FCR compared to the Std ME, High ME, and Choice feeding groups ($p < 0.05$). In the CON experiment, *ad libitum*-fed pullets had greater FCR than restricted-fed birds ($p < 0.05$), whereas the lower the dietary ME the greater the FCR (Low ME > Std ME > High ME; $p < 0.05$). Although simplistic, birds tend to eat to satisfy their energy requirements [50], therefore, higher ME diets supply the same amount of energy at a lower intake and consequently lower FCR. These results were in accordance with previous trials that reported increased FCR with decreased dietary energy [34,37,50,51]. However, while similar BW can be achieved with different dietary ME levels, body composition might differ, especially when the ratio of other nutrients (crude protein and AA) is kept constant across the different dietary energy levels. Similarly to dietary energy, feed restriction tends to show higher feed efficiency due to decreased maintenance requirements and fat deposition [11,52]. Although no maintenance requirements and fat deposition results are presented in the present report, the greater feed efficiency in the restricted-fed compared to *ad libitum*-fed birds corroborates the results of the present study.

**Table 8.** Cumulative feed conversion ratio (FCR, g of feed:g of BW gain) at 18 wks of age of Lohmann Brown-Lite pullets using different feed allocation (FA) and dietary metabolizable energy levels (Diet ME) in the precision (PF) and conventional (CON) experiments.

| Effect | | PF Experiment [1] | | CON Experiment [2] | |
|---|---|---|---|---|---|
| | | FCR | | FCR | |
| | | Mean | SEM | Mean | SEM |
| | | g:g | | | |
| FA | Restricted | 4.167 | 0.028 | 4.464 [b] | 0.037 |
| | MEV or *ad libitum* | 4.228 | 0.049 | 4.635 [a] | 0.020 |
| Diet ME | Low ME | 4.483 [a] | 0.053 | 4.846 [a] | 0.020 |
| | Std ME | 4.178 [b] | 0.082 | 4.585 [b] | 0.011 |
| | High ME | 3.994 [b] | 0.037 | 4.218 [c] | 0.059 |
| | Choice | 4.136 [b] | 0.043 | - | - |
| FA × Diet ME | Low ME × Restricted | 4.498 | 0.038 | 4.741 | 0.037 |
| | Std ME × Restricted | 4.143 | 0.066 | 4.464 | 0.015 |
| | High ME × Restricted | 3.959 | 0.037 | 4.187 | 0.105 |
| | Choice × Restricted | 4.070 | 0.074 | - | - |
| | Low ME × MEV or *ad libitum* | 4.469 | 0.098 | 4.951 | 0.016 |
| | Std ME × MEV or *ad libitum* | 4.212 | 0.151 | 4.706 | 0.017 |
| | High ME × MEV or *ad libitum* | 4.030 | 0.065 | 4.249 | 0.055 |
| | Choice × MEV | 4.202 | 0.042 | - | - |
| | | *p*-value | | | |
| FA | | 0.28 | | 0.007 | |
| Diet ME | | <0.001 | | <0.001 | |
| FA × Diet ME | | 0.70 | | 0.35 | |

[a–c] Means within columns with no common subscript differ ($p < 0.05$). [1] Feed allocation (FA) at two levels: meal every visit (MEV) and restricted to the lower boundary of the Lohmann Brown-Lite recommended target BW trajectory. Dietary apparent metabolizable energy treatments (Diet ME): Low (2600 kcal/kg), Standard (Std; 2800 kcal/kg), or High ME (3000 kcal/kg). Choice treatment enabled birds to choose from the three diets. [2] Feed allocation (FA) at two levels: *ad libitum* and restricted to the lower boundary of the Lohmann Brown-Lite recommended target BW trajectory. Dietary apparent metabolizable energy treatments (Diet ME): Low (2600 kcal/kg), Standard (Std; 2800 kcal/kg), or High ME (3000 kcal/kg).

### 3.4. Feeding Motivation

Feeding motivation was studied only in the PF experiment (Table 9). Increased number of daily visits to the feeding stations, and decreased daily meals, meal size, and successful station visits were observed in restricted-fed pullets ($p < 0.05$). Pullets fed High ME had a lower number of visits and number of meals per day and a greater meal size compared to the other treatment groups ($p < 0.05$). The interaction between the FA and dietary ME showed a greater number of visits in the Choice restricted group when compared to Std and High ME, but this difference was not observed when birds were fed at every visit ($p < 0.05$).

The High ME-fed group had the lowest number of meals when pullets were fed at MEV whereas when restricted-fed, the High ME pullets had a lower number of meals than Low and Std ME only ($p < 0.05$). Additionally, no difference was observed between meal size when pullets were feed restricted, whereas in the MEV treatment High ME pullets had greater meal size than each of the other groups ($p < 0.05$).

**Table 9.** Feeding station visit frequency, meal frequency, meal size and successful rate of visits of Lohmann Brown-Lite pullets fed different feed allocation (FA) and dietary metabolizable energy levels (Diet ME) in the precision feeding (PF) experiment, 6 to 18 wk.

| Effect | | PF Experiment [1] | | | | | | | |
|---|---|---|---|---|---|---|---|---|---|
| | | Visits | SEM | Meals | SEM | Meal Size | SEM | Successful Visits [2] | SEM |
| | | —— n —— | | —— n —— | | —— g —— | | —— % —— | |
| FA | Restricted | 9.3 a | 0.40 | 4.2 b | 0.11 | 7.8 b | 0.38 | 62.4 b | 2.6 |
| | MEV | 5.9 b | 0.22 | 5.6 a | 0.16 | 10.3 a | 0.29 | 95.0 a | 0.8 |
| Diet ME | Low ME | 8.2 a | 0.54 | 5.5 a | 0.16 | 8.5 b | 0.35 | 79.6 | 3.1 |
| | Std ME | 7.7 a | 0.40 | 5.7 a | 0.25 | 8.3 b | 0.56 | 81.9 | 2.7 |
| | High ME | 5.8 b | 0.50 | 3.7 c | 0.15 | 11.1 a | 0.59 | 78.8 | 3.4 |
| | Choice | 8.7 a | 0.36 | 4.7 b | 0.21 | 8.2 b | 0.34 | 74.3 | 1.3 |
| Period (wk) | 6 to 8 | 6.5 b | 0.30 | 4.6 | 0.17 | 7.5 b | 0.33 | 81.9 | 1.5 |
| | 9 to 11 | 7.9 a | 0.42 | 5.2 | 0.19 | 8.8 ab | 0.44 | 79.6 | 3.2 |
| | 12 to 14 | 8.3 a | 0.49 | 5.1 | 0.22 | 9.6 a | 0.52 | 75.7 | 3.1 |
| | 15 to 18 | 7.7 ab | 0.57 | 4.8 | 0.20 | 10.3 a | 0.58 | 77.4 | 2.9 |
| FA × Diet ME | Low ME × Restricted | 10.1 ab | 1.01 | 5.0 b | 0.22 | 7.4 c | 0.56 | 64.2 | 5.9 |
| | Std ME × Restricted | 8.1 bc | 0.64 | 4.4 bc | 0.25 | 8.3 bc | 0.72 | 69.7 | 5.2 |
| | High ME × Restricted | 7.5 bc | 0.91 | 3.4 d | 0.20 | 9.0 bc | 1.09 | 62.2 | 6.7 |
| | Choice × Restricted | 11.5 a | 0.57 | 3.9 cd | 0.21 | 6.6 c | 0.56 | 53.4 | 2.0 |
| | Low ME × MEV | 6.4 c | 0.40 | 6.1 a | 0.22 | 9.7 b | 0.44 | 95.1 | 1.6 |
| | Std ME × MEV | 7.3 bc | 0.49 | 6.9 a | 0.44 | 8.3 bc | 0.87 | 94.2 | 1.6 |
| | High ME × MEV | 4.1 d | 0.42 | 3.9 cd | 0.21 | 13.3 a | 0.44 | 95.4 | 1.7 |
| | Choice × MEV | 5.8 cd | 0.45 | 5.6 ab | 0.37 | 9.8 b | 0.40 | 95.2 | 1.8 |
| | | —————————— *p*-value —————————— | | | | | | | |
| FA | | <0.001 | | <0.001 | | <0.001 | | <0.001 | |
| Diet ME | | <0.001 | | <0.001 | | <0.001 | | 0.042 | |
| Period | | 0.005 | | 0.080 | | <0.001 | | 0.22 | |
| FA × Diet ME | | <0.001 | | 0.002 | | 0.033 | | 0.060 | |
| FA × Period | | 0.31 | | 0.098 | | 0.34 | | <0.001 | |
| Diet ME × Period | | 0.92 | | 0.88 | | 0.93 | | 0.63 | |
| FA × Diet ME × Period | | 0.96 | | 0.99 | | 0.98 | | 0.89 | |

a–d Means within columns with no common subscript differ ($p < 0.05$). [1] Feed allocation (FA) at two levels: meal every visit (MEV) and restricted to the lower boundary of the Lohmann Brown-Lite recommended target BW trajectory. Choice treatment enabled birds to choose from the three dietary apparent metabolizable energy levels: Low (2600 kcal/kg), Standard (Std; 2800 kcal/kg), or High ME (3000 kcal/kg). [2] Percentage rate calculated as the daily meals divided by daily visits and the result multiplied by 100.

Controlling the FI of pullets based on a target BW imposed a constraint factor to the restricted-fed pullets and this restriction factor increased feeding motivation in laying hens [53]. Since 95% of the visits from the MEV group were successful, it is possible that MEV birds had lower feed motivation (number of visits) because they were able to more closely meet their nutrient intake requirements compared to restricted-fed pullets. Moreover, the greater number of meals in the MEV group, despite the greater visits in the restricted group, was reflected in the lower success rate of visits in the restricted-fed group (62.4%). Afrouziyeh et al. [54], using a first-generation PF system with broiler breeders, found a slight decrease in the feeding motivation index (daily station visits:meal ratio) due to feed restriction relaxation. Feed motivation index was not calculated in the current experiment since the increased visit number in the restricted-fed birds was clearly observed (63% greater). Interestingly, High ME-fed birds had lower visit numbers and meal numbers, explained by the greater meal size in the High ME × MEV group. Feed density is a possible explanation for the greater meal size in this group, where the same amount of feed volume consumed can yield greater weights in a more concentrated diet. On the other hand, diet

dilution has also been reported as a potential factor in reducing feed motivation in broiler breeders [55]. However, this dilution can decrease nutrient intake, and it seems that energy dilution did not affect layer pullet feeding motivation in the current study.

## 4. Conclusions

The current trial was the first utilizing a precision feeding system equipped with multiple feeders, having the capacity for feeding specific diets to individual free-run birds. The results of the current study indicate that birds fed MEV and *ad libitum* increased BW, ADFI and MEI compared to birds whose BW was restricted to the lower boundary of the Lohmann-Brown-Lite recommended BW trajectory. The greatest BW uniformity was observed in the restricted precision-fed pullets. Pullets fed lower dietary ME levels increased ADFI and FCR and were not able to maintain similar MEI. Additionally, when given a choice, pullets preferred to consume diets containing greater ME levels.

Based on efficiency, feeding birds with greater dietary ME and feed restriction is recommended, whereas for maximum growth (BW) and uniformity the feed restriction is not recommended. Despite the current results, better recommendations should be provided for after the laying cycle. The next steps of this project will be examining reproductive performance and carcass composition of the pullets reared in the current trial, and more precise recommendations will be made based on dietary energy and feed restriction.

**Author Contributions:** Conceptualization, T.L.N., R.P.K., M.J.Z. and L.S.; methodology, T.L.N., D.R.K., M.J.Z., R.P.K. and L.S.; software, T.L.N., J.A.C. and M.J.Z.; validation, T.L.N., J.A.C., D.R.K. and M.J.Z.; formal analysis, T.L.N.; investigation, T.L.N., J.A.C. and M.J.Z.; resources, D.R.K. and M.J.Z.; data curation, T.L.N., J.A.C. and M.J.Z.; writing—original draft preparation, T.L.N.; writing—review and editing, T.L.N., J.A.C., D.R.K., M.J.Z., R.P.K. and L.S.; visualization, T.L.N.; supervision, M.J.Z.; project administration, T.L.N. and M.J.Z.; funding acquisition, M.J.Z. All authors have read and agreed to the published version of the manuscript.

**Funding:** Financial support from Alberta Agriculture and Forestry (Edmonton, AB, Canada), Egg Farmers of Alberta (Calgary, AB, Canada), and Egg Farmers of Canada (Ottawa, ON, Canada) is gratefully acknowledged. This research was also supported by the public–private partnership project 'Customized nutrition in layer pullets' (LWV21.115), which included financial support from the Ministry of Agriculture, Nature, and Food Quality (The Netherlands), ForFarmers Nederland B.V., ABZ De Samenwerking B.V., AgruniekRijnvallei Voer B.V., Cargill Animal Nutrition, Nutreco Servicios SA, DSM/Twilmij B.V., Vitelia Voeders B.V., Institut de Sélection Animale B.V., Precision ZX Inc., Poultry Expertise Centre, and collaborative input from collaborating knowledge institutes the University of Alberta, Wageningen University and Schothorst Feed Research and Aeres University of Applied Sciences.

**Institutional Review Board Statement:** The animal study protocol was approved by the Institutional Animal Care Use Committee Livestook of University of Alberta (AUP00003662, approved on 22 January 2020).

**Informed Consent Statement:** Not applicable.

**Data Availability Statement:** Data available upon request.

**Acknowledgments:** In kind development, manufacturing, and technical support for the precision feeding system was provided by Xanantec Technologies, Inc. (Edmonton, AB, Canada). The exceptional technical expertise provided by staff and students at the University of Alberta Poultry Research Centre (Edmonton, AB, Canada) is gratefully acknowledged.

**Conflicts of Interest:** The authors declare no conflict of interest.

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
