# Peer review of "Effects of Metabolizable Energy Intake and Body-Weight Restriction on Layer Pullets: 1-Growth, Uniformity, and Efficiency"

_poultry, doi:10.3390/poultry2040036_

Round 1

Reviewer 1 Report

Comments and Suggestions for Authors

Overall the manuscript is difficult to follow and with many variables measured with the complexity of the proposed model is very difficult to follow and understand what is what the authors are evaluating. Sometimes I was not sure if you were discussing Experiment 1 or Experiment 2. Could you separate and have 2 manuscripts instead? Could this clean the data and be more concise? 

Additional comments in PDF file

Comments on the Quality of English Language

No comments on this section

Author Response

Dear reviewer,

We are thankful for your time to review our manuscript (poultry-2647676).

Below are the responses for your suggestions and requests:

Overall the manuscript is difficult to follow and with many variables measured with the complexity of the proposed model is very difficult to follow and understand what is what the authors are evaluating. Sometimes I was not sure if you were discussing Experiment 1 or Experiment 2. Could you separate and have 2 manuscripts instead? Could this clean the data and be more concise? 

Response: Thanks for your comments and suggestions, we appreciate your time and consideration. We (the authors) think that having both trials in the same manuscript is important for the complete understanding of the results. We had a limited number of precision feeding stations and, therefore a limited number of birds per treatment group (23 birds). Thus, the conventional trial was additional to the precision feeding in order to have extra birds for dissection and body composition analysis (L96 to 97 and L118 to 120). The story is more complete when using both trials despite the amount of data. Improvements were made throughout the text for better understanding of the trial (highlighted yellow) based on your suggestions.

PDF file attached:

If this Feed Intake????

By restricting feed intake which regulates BW.

Response: Thank you sincerely for your suggestion. We use feed restriction to control the body weight and we considered using “feed restriction” instead of “body weight restriction” for the title. However, our birds were fed to a specific body weight, so we believe that body weight restriction would be correct.

Are these results from Experiment 1??? Experiment 2??? Combined??? This section needs clarification to know what are you referencing.

Response: Additional descriptions were included for the results reported (L27 and L31).

check this sentence it does not sounds correct...

Response: Sentences in L51 to 52 were modified.

Can you explain this further??? Is your experimental unit still the bird???

Response: Thank you for the question. More details were presented in L116.

For experiment 1???? Experiment 2???

Please clarify- and add experiment 1 or 2 in the title so we know what you are talking about

Response: Additional explanation was added to the title of Table 1. Thank you.

What is the () refering to?

Response: The values between parentheses are analyzed results. Showed in the “Nutrient calculated (analyzed)” section in the table; added in L133 under Table 1.

The data were also collected by RFID system?? from individual birds ??? How can this be comparable to the previous with the basis of taking individual bird as experimental unit???

Response: The data collection description is in the section “Data collection”. Data collection was individually recorded for BW and used for CV calculations as well (each bird identified with an RFID in both PF and CON trials, L146 to 147). However, the other variables (FCR and MEI) were calculated at the pen level (L208 and 209).

Is this correct???

How can you run stats on a trial with 2 replicates.

Birds were fed by pen, so the pen should be your experimental unit. How are you confident about your variation by pen with only 2 pens???

Response: We are aware of the limitations of the CON trial; its main purpose was to have extra birds for dissection, so birds would not be removed from the PF system. Although the treatments were applied to the pen level, analysis of BW could be performed with individual birds.

Did your removed non-significant interactions from your final model???

Response: No, all interactions probabilities were presented, whether significant or not.

But you have 2 factors. So age is an extra factor =, right???  Would this then be a repeated measures analysis?????

Response: Yes, age was treated as a factor in the model for variables observed over time. Repeated measurements could also be performed in this case, where age would be a repeated factor in the model. There are advantages and disadvantages to both methods, but mostly, the objective of using time as a factor in the model is to assess the changes due to time instead of accounting for the variance in the repeated model. Age as a factor in this case was not analyze at specific age (e.g., one analysis at week 6 and another at week 18) but in the model.

Thank you for your revision!

Sincerely.

Reviewer 2 Report

Comments and Suggestions for Authors

This study investigated the effects of metabolizable energy intake and body weight restriction on layer pullets using both precision feeding and conventional feeding systems. The findings of this study suggest that precision feeding systems equipped with multiple feeders can be used to improve the growth and development of layer pullets. The data collection method utilized in this study is generally well-known, and the experimental design is reasonable. The manuscript is well written. With minor corrections, it can be considered for publication in Poultry.

1.       Some error was shown throughout the entire manuscript (eg. Line 124), which made it hard to locate all of the tables.

2.        Typo: Lines 390 and 391, only brackets should be used for citations. 

Author Response

Dear reviewer,

We are thankful for the review and suggestions for our manuscript (poultry-2647676).

Below are the responses for your suggestions and requests:

This study investigated the effects of metabolizable energy intake and body weight restriction on layer pullets using both precision feeding and conventional feeding systems. The findings of this study suggest that precision feeding systems equipped with multiple feeders can be used to improve the growth and development of layer pullets. The data collection method utilized in this study is generally well-known, and the experimental design is reasonable. The manuscript is well written. With minor corrections, it can be considered for publication in Poultry.

1.       Some error was shown throughout the entire manuscript (eg. Line 124), which made it hard to locate all of the tables.
Response: Errors for the tables citation have been fixed, and hopefully, it should be showing correctly now. Thank you.

  1.        Typo: Lines 390 and 391, only brackets should be used for citations.

Response: Modification for references made in L384 and 385

Thank you for the suggestions!

Sincerely.

Reviewer 3 Report

Comments and Suggestions for Authors

Dear authors, 

The manuscript was well-written and the content was informative and well-presented. I commend the authors for the comprehensive and systematic review of the topic. The manuscript will be a valuable contribution to this journal.

However, I’ve mentioned a few minor comments in the comment section of the main manuscript file that need to be addressed before the manuscript can be published. Some of these are the following:

Please revise your title and try to use more appropriate eye-catching words for the title of this study. 

Please add one line at the end of the abstract, which basically explains the basic output of this study and the future recommendations related to this study work as well.

Line 88: Does FCR decrease or increase as a result of more dietary feeding?

Line 124: Please cite an appropriate reference here

Line 127: Please cite an appropriate reference here

Line 244: Please cite an appropriate reference here

Line 251: Please cite an appropriate reference here

Line 286, 287, 288: Please cite an appropriate reference here

Line 495: Please add some future recommendations based on your current findings in the conclusion section. 

Line 529: Please follow the Journal guidelines for reference management. 

Best wishes

Author Response

Dear reviewer,

We are thankful for the review and suggestions for our manuscript (poultry-2647676).

Below are the responses for your suggestions and requests:

Dear authors,
The manuscript was well-written and the content was informative and well-presented. I commend the authors for the comprehensive and systematic review of the topic. The manuscript will be a valuable contribution to this journal.
However, I’ve mentioned a few minor comments in the comment section of the main manuscript file that need to be addressed before the manuscript can be published. Some of these are the following:
Please revise your title and try to use more appropriate eye-catching words for the title of this study.

Response: Because this manuscript is a part of a larger project, several publications will be coming after this, which will be numbered (2, 3, 4, etc.). This is the reason why we would like to keep the main title despite its simplicity (Effects of Metabolizable Energy Intake and Body Weight Restriction on Layer Pullets).

Please add one line at the end of the abstract, which basically explains the basic output of this study and the future recommendations related to this study work as well.
Response: Thank you for the suggestion. Additional sentence was added to the abstract, L35 to L38.

Line 88: Does FCR decrease or increase as a result of more dietary feeding?
Response: The FCR would decrease with increase energy in the diet, whereas FCR would increase with the increase feed intake (dietary feeding). If we use feed efficiency that would be the inverse expectation.

Line 124: Please cite an appropriate reference here
Line 127: Please cite an appropriate reference here
Line 244: Please cite an appropriate reference here
Line 251: Please cite an appropriate reference here
Line 286, 287, 288: Please cite an appropriate reference here
Response: All the references were corrected and are available in the new manuscript version.

Line 495: Please add some future recommendations based on your current findings in the conclusion section.
Response: A recommendation section was included at the end of conclusion.

Line 529: Please follow the Journal guidelines for reference management.
Response: References were updated as per the journal guidelines.

Thank you in advance!

Sincerely.

Reviewer 4 Report

Comments and Suggestions for Authors

The work focuses on effects of metabolizable energy intake and body weight restriction on layer production. The manuscript is generally well-written. However, there are some aspects mentioned below that must be reviewed and corrected, since today they would prevent its acceptance for publication. Please, consider all the recommendations that I describe below.

1.     Line 14: The aim of the study needs to be clearly addressed in the abstract.

2.     The conclusion and recommendation require in the termination of abstract (Line 33) and in the conclusion section (Line 495).

3.      Please check the citied references through the manuscript, mostly including error

4.      Table 1, please add the ingredients in standard diets. Recheck the chemical composition especially Canola oil levels among different diets.

5.     Why did you use High mount of wheat in the diets?. Wheat includes starches, which are difficult to digest in the foregut. These carbohydrates provide food for lower gut bacteria, which can result in waxy particles and sticky feces. How did you deal with these issues?

6.     What were the experimental units for the various parameters in statistical analysis?

7.     What are the underlying mechanisms that explain differences in growth performance in layer pullets dependent on dietary energy and feed restriction?  Must be specified in the discussion section. You did not explain the underlying causes of these changes.    

Author Response

Dear reviewer,

We are thankful for the review and suggestions for our manuscript (poultry-2647676).

Below are the responses for your suggestions and requests:

The work focuses on effects of metabolizable energy intake and body weight restriction on layer production. The manuscript is generally well-written. However, there are some aspects mentioned below that must be reviewed and corrected, since today they would prevent its acceptance for publication. Please, consider all the recommendations that I describe below.

  1. Line 14: The aim of the study needs to be clearly addressed in the abstract.

Response: Additional information added in L15.

  1. The conclusion and recommendation require in the termination of abstract (Line 33) and in the conclusion section (Line 495).

Response: Additional information added to the end of abstract and conclusion section. Thank you.

  1. Please check the citied references through the manuscript, mostly including error

Response: References were updated in the manuscript and errors removed.

  1. Table 1, please add the ingredients in standard diets. Recheck the chemical composition especially Canola oil levels among different diets.

Response: The standard diet was developed from the mixture of 1:1 of the low and high ME diet, so ingredients are not presented. Ingredients were checked again, and no changes were required.

  1. Why did you use High mount of wheat in the diets?. Wheat includes starches, which are difficult to digest in the foregut. These carbohydrates provide food for lower gut bacteria, which can result in waxy particles and sticky feces. How did you deal with these issues?

Response: Thank you for this question. Wheat is the main ingredient for poultry in Canada and common in Europe (20 to 40% of the diet, depending on the region), so availability and relevance to our primary stakeholders was the main reason. We mitigate these concerns by using enzymes in the diets (xylanase mainly, but also phytase; Table 1).

  1. What were the experimental units for the various parameters in statistical analysis?

Response: In the precision feeding system, the experimental unit was the bird for all parameters (L226), whereas in the CON system the pen was the experimental unit for feed intake, MEI, and FCR, and the individual bird was used for BW and CV (L229 to 230).

  1. What are the underlying mechanisms that explain differences in growth performance in layer pullets dependent on dietary energy and feed restriction?  Must be specified in the discussion section. You did not explain the underlying causes of these changes.

Response: Thanks for this question. We did not find underlying growth performance differences (BW) due to dietary energy since birds tend to eat to their energy requirements. Regarding the feed restriction factor, the underlying causes are basically nutrient availability for growth (which is discussed in the other sections). We acknowledge that not a lot of discussion is presented in the BW section, however, additional discussion was provided in the following sections (Feed and Metabolizable Energy Intake and Feed Conversion Ratio).

We are thankful for your suggestions and comments!

Sincerely,

Round 2

Reviewer 1 Report

Comments and Suggestions for Authors

Thank you for your effort addressing comments